# Polymorphisms in *ERCC5* rs17655 and *ERCC1* rs735482 Genes Associated with the Survival of Male Patients with Postoperative Oral Squamous Cell Carcinoma Treated with Adjuvant Concurrent Chemoradiotherapy

**DOI:** 10.3390/jcm8010033

**Published:** 2019-01-01

**Authors:** Thomas Senghore, Huei-Tzu Chien, Wen-Chang Wang, You-Xin Chen, Chi-Kuang Young, Shiang-Fu Huang, Chih-Ching Yeh

**Affiliations:** 1School of Public Health, College of Public Health, Taipei Medical University, Taipei 11031, Taiwan; tsenghore@gmail.com (T.S.); youxin810@gmail.com (Y.-X.C.); 2Department of Nursing, School of Medicine and Allied Health Sciences, University of The Gambia, Independence Drive, Banjul, P.O. Box 1646, The Gambia; 3Department of Public Health, Chang Gung University, Tao-Yuan 33305, Taiwan; kathy.htchien@gmail.com; 4Department of Nutrition and Health Sciences, Chang Gung University of Science and Technology, Taoyuan 33302, Taiwan; 5Ph.D. Program for Translational Medicine, College of Medical Science and Technology, Taipei Medical University, Taipei 11031, Taiwan; wangwc@tmu.edu.tw; 6Department of Otolaryngology, Chang Gung Memorial Hospital, Keelung 20401, Taiwan; rioriorioman@gmail.com; 7Department of Otolaryngology, Head and Neck Surgery, Chang Gung Memorial Hospital, Linkou, Taoyuan 33305, Taiwan; 8Department of Public Health, College of Public Health, China Medical University, Taichung 40402, Taiwan

**Keywords:** nucleotide excision repair, genetic polymorphism, oral squamous cell carcinoma, concurrent chemoradiotherapy, prognosis

## Abstract

The nucleotide excision repair (NER) pathway plays a major role in the repair of DNA damaged by exogenous agents, such as chemotherapeutic and radiotherapeutic agents. Thus, we investigated the association between key potentially functional single nucleotide polymorphisms (SNPs) in the NER pathway and clinical outcomes in oral squamous cell carcinoma (OSCC) patients treated with concurrent chemoradiotherapy (CCRT). Thirteen SNPs in five key NER genes were genotyped in 319 male OSCC patients using iPLEX MassARRAY. Cox proportional hazards models and Kaplan–Meier survival curves were used to estimate the risk of death or recurrence. Carriers of the *XPC* rs2228000 TT genotype showed a borderline significant increased risk of poor overall survival under the recessive model (hazard ratio (HR) = 1.81, 95% confidence interval (CI) = 0.99–3.29). The CC genotypes of *ERCC5* rs17655 (HR = 1.54, 95% CI = 1.03–2.29) and *ERCC1* rs735482 (HR = 1.65, 95% CI = 1.06–2.58) were associated with an increased risk of worse disease-free survival under the recessive model. In addition, participants carrying both the CC genotypes of *ERCC5* rs17655 and *ERCC1* rs735482 exhibited an enhanced susceptibility for recurrence (HR = 2.60, 95% CI = 1.11–6.09). However, no statistically significant interaction was observed between them. Our findings reveal that the *ERCC5* rs17655 CC and *ERCC1* rs735482 CC genotypes were associated with an increased risk of recurrence in male patients with OSCC treated with CCRT. Therefore, CCRT may not be beneficial, and alternative treatments are required for such patients.

## 1. Introduction

Oral squamous cell carcinoma (OSCC) is the leading cause of cancer morbidity and mortality, especially among men in Taiwan [1]. Despite new advances in the diagnosis and therapeutic approaches, the 5-year survival remains low [1,2]. While relapse of OSCC remains a major clinical challenge, the incidence of relapse among patients varies, even for those with a similar stage of disease at diagnosis or those who undergo the same treatment [3]. This implies that other factors, such as genetic variations, may play an important role in disease prognosis.

Most patients with OSCC are diagnosed at an advanced stage of the disease [4]. For these patients, the treatment options are limited to mainly systemic therapy, often as concurrent chemoradiotherapy (CCRT) with platinum-based DNA damaging agents as either primary treatment or adjuvant postoperative therapy [5,6,7]. However, the overall survival (OS) for these patients remains poor because most of them experience recurrence or distance metastases [8,9,10]. Genetic variations in DNA repair genes affect susceptibility to the efficacy and survival outcome of a certain treatment [11,12]. Increased DNA repair capacity may affect the sensitivity of the tumor cells to chemotherapy and radiotherapy (RT) by allowing cancer cells to repair DNA that has been damaged by these agents. Single nucleotide polymorphism (SNP) in genes involved in the nucleotide excision repair (NER) pathway may modulate DNA repair capacity by influencing gene expression or activity, thereby affecting the anticancer effects of therapeutic agents and treatment response [13,14].

The excision repair cross-complementation genes, including groups 1 (*ERCC1*), 2 (*ERCC2*), and 5 (*ERCC5*) and xeroderma pigmentosium complementation group A (*XPA*) and C (*XPC*) encode proteins that are involved in the NER pathway; and together with other proteins, operate to recognize and repair damaged DNA [15]. The XPC together with XPF initially recognize the DNA lesion that is unwound and remodeled by helicase proteins ERCC3 and ERCC2 that binds to XPA and replication protein A (RPA). The ERCC1 and ERCC5 proteins are involved in the incision of the identified DNA lesion. The difference in treatment response and clinical outcome have been attributed to SNPs in genes that code of the above proteins [13,14]. Therefore, identifying genetic markers in the NER pathway may help develop personalized management strategies, thereby maximizing treatment success and improving survival.

Thus, we conducted a retrospective cohort study to test whether SNPs in genes involved in the NER pathway are associated with prognosis in male patients with OSCC treated with adjuvant CCRT. A total of 13 SNPs in *ERCC5*, *ERCC2*, *ERCC1*, *XPC*, and *XPA* genes, which have been found to affect the risk and/or survival of cancers, were selected in the present study [13,14,16,17,18,19,20]. Their associations with clinical outcomes were evaluated using alternative genetic models, including additive, dominant, and recessive models.

## 2. Material and Methods

### 2.1. Study Population

In total, 360 male participants newly diagnosed with histopathological confirmed advanced OSCC who received surgery plus adjuvant CCRT were recruited from the Head and Neck Surgery Department’s Cancer Registry at Chang Gung Memorial Hospital, LinKou, Taiwan, from 1999 to 2016. A total of 41 participants were excluded, including 13 of aboriginal ethnicity, 23 with early-stage oral cancer (TNM stages I and II), and 5 with missing information on clinicopathologic variables (TNM stage, vascular invasion, and extracapsular spread). A final sample of 319 was included for analysis. Information on demographic characteristics (age, education, occupation, and ethnicity), lifestyle habit (cigarette smoking, alcohol drinking, and betel nut chewing), and family cancer history were collected through an interviewer-administered questionnaire. Lifestyle habits were categorized as either never (if the person never engaged in the habit continuously for more than a year) or ever (if the person ever engaged in the habit for more than a year). From the weight and height measurements, body mass index (BMI) was calculated as weight/height^2^ (kg/m^2^). Clinical information was also collected before treatment through a detailed medical history, physical examination, completed blood count, routine blood chemistry, computed tomography (CT) or magnetic resonance imaging (MRI) of the head and neck, abdominal ultrasound, and whole body bone scan or positron emission tomography scan. This study was approved by the Chang Gung Memorial Hospital (IRB No. 201800213B0) and Taipei Medical University ethics review committees (IRB No. N201802083). All participants provided written informed consent after a detailed explanation of study objectives.

### 2.2. Sample Preparation and DNA Extraction

For each participant, a pair of tumor and normal adjacent nontumor tissue samples were surgically removed, dissected into small pieces, and immediately stored in liquid nitrogen at −80 °C. The surgically removed samples were then sent for pathological examination and staging as per the seventh edition of the American Joint Committee on Cancer—TNM staging system [21]. Histology diagnosis was defined as squamous cell carcinoma, verrucous carcinoma, cylindric cell carcinoma, adenoid cystic carcinoma, mucoepidermoid carcinoma, and adenocarcinoma. For this study, only those with a diagnosis of squamous cell carcinoma were included. Venous blood samples were also collected and stored in heparinized tubes. Germline DNA was extracted from buffer-coated cells using the standard phenol-chloroform method and prepared for genotyping.

### 2.3. SNP Selection and Genotyping

SNPs in the NER pathway were selected from studies that indicated that SNPs were associated with the risk or prognosis of malignancies in ethnic Chinese [16,18,19,20]. A total of 13 potentially functional SNPs in *ERCC5* (rs2094258, rs1047768, rs17655, and rs873601), *ERCC2* (rs13181 and rs1799793), *ERCC1* (rs735482, rs3212986, and rs11615), *XPC* (rs2228001 and rs2228000), and *XPA* (rs1800975 and rs10817938) genes were genotyped using the Sequenom iPLEX MassARRAY system (Sequenom, Inc., San Diego, CA, USA). A 10% random sample was reanalyzed, and it showed 100% concordance for all the polymorphisms.

### 2.4. Patient Treatment and Follow-Up

All patients underwent radical tumor excision with clinical stage-based neck dissection after preoperative tumor survey. The primary tumors were resected above 1-cm safety margins (both peripheral and deep margins). Neck dissections were performed according to examination status. If the lesion invaded deeply and crossed the midline, as observed in tongue cancer, bilateral neck dissection was performed. Pathologic parameters included tumor stage, nodal status, extranodal extension (ENE), tumor cell differentiation, perineural invasion, skin invasion, bone invasion, and tumor depth. Postoperative RT was administered to patients with pT4 stage tumor, pathologically close margins (≤4 mm), or pathologically positive lymph nodes. The radiation dose lay between 6000 and 6600 cGy. CCRT with cisplatin-based agents was administered to patients with ENE or pathological multiple lymph node metastases 4 to 8 weeks after the surgical procedure. During the course of RT, 5-fluorouracil was administered orally.

Following commencement of treatment, the participants were monitored during their treatment and after treatment through regular clinical and radiological examinations. Follow-up involved monthly checkups for the first 6 months. This was followed by checkups every 2 months in the second 6 months, then checkups every 3 months within the second year, and checkups every 6 months thereafter. The follow-up included an analysis of medical history, physical examination (including complete oral examination), laboratory examination, X-rays, and CT or MRI. To confirm recurrence, histology of biopsy or imaging studies were conducted. Data for all deaths resulting from OSCC were based on death certificates.

### 2.5. Statistical Analysis

Statistical analysis was performed using SAS (version 9.4 for Windows; SAS institute, Cary, NC, USA). Demographic and clinical characteristics were summarized as mean and standard deviation for continuous variables and frequency and proportions for categorical variables. The distribution of genotypes by clinical characteristics was assessed using Chi-square test. Major clinical outcomes were disease-free survival (DFS) and OS. DFS was measured from the first day of treatment to the time of recurrence, metastasis, or death due to any cause. OS was calculated as the time elapsed (in months) from the date of commencing RT to the date of death. Patients without an event at the date of the last contact were considered as being censured or subject to administrative censoring by the end of the follow-up period. Survival analysis was conducted using the Kaplan–Meier method, and survival curve differences among the genotypes were compared using the log-rank test. Univariate and multivariate Cox proportional hazards models were used to evaluate the effects of demographic, clinical characteristics, and SNPs on survival. Hazard ratios (HRs) and their 95% confidence intervals (CIs) were used to estimate the relative risk of death or recurrence. We evaluated the individual variants in different genetic models, including additive, dominant, and recessive models, on OSCC survival. Sociodemographic and clinical factors significant in the univariate analysis were adjusted in multivariate Cox regression models. Furthermore, multiplicative interactions were evaluated using the likelihood ratio test. Due to the location of multiple SNPs within the same chromosome or gene, linkage disequilibrium (LD) analysis was performed using Haploview (version 4.2). For those SNPs within the same block that were found to be in high LD with each other, further haplotypes analysis was performed using PHASE software (version 2.1) [22]. Statistical significance was set at *p* < 0.05 and was two-sided.

## 3. Results

### 3.1. Demographic and Clinical Characteristics of Study Participants

The demographic and clinical characteristics of the study patients are summarized in Table 1. The mean age was 49.72 ± 9.8. Most participants were under 50 years (51.41%) old, of Taiwanese descent (paternal 72.10% and maternal 74.92%), had normal BMI (49.22%), had smoked cigarettes (according to the “ever” criterion; 85.25%), drank alcohol (69.28%), and chewed betel nut (86.21%). A considerable number of patients exhibited poor clinical characteristics. In total, 55 (17.24%) had poorly differentiated tumors, 197 (61.76%) had primary tumor size in the T3 to T4 range, 217 (67.92%) had N2 to N3 nodal involvement, 177 (55.49%) had perineural invasion, 19 (5.96%) had vascular invasion, 40 (12.54%) had lymphatic invasion, 205 (64.24%) had ENE, and 277 (86.83%) had pathologic TNM stage IV. The genotype frequency distribution analysis showed a statistically significant difference in genotypes of *ERCC1* rs11615 in terms of tumor differentiation (*p* = 0.039), *XPC* rs2228000 in terms of vascular invasion (*p* = 0.045), *ERCC1* rs3212986 and *XPA* rs10817938 in terms of lymphatic invasion (*p* = 0.046 and 0.033, respectively), *XPC* rs2228001 in terms of pathologic TNM stage (*p* = 0.039), and *ERCC5* rs17655 in terms of DFS (*p* = 0.049) (Appendix A).

### 3.2. Survival Analysis

The median (range) follow-up duration was 15 months (1–199 months) and 12 months (1–199 months) for OS and DFS, respectively. In the univariate analysis, N2–N3 nodal involvement (HR = 2.41, 95% CI = 1.42–4.10), lymphatic invasion (HR = 2.18, 95% CI = 1.30–3.67), and ENE (HR = 3.91, 95% CI = 2.13–7.19) were significantly associated with OS, whereas primary tumor size in the range of T3 to T4 (HR = 1.72, 95% CI = 1.17–2.53), N2–N3 nodal involvement (HR = 1.63, 95% CI = 1.09–2.44), and ENE (HR = 1.79, 95% CI = 1.20–2.69) were significantly associated with DFS. However, no significant association was observed between demographic and lifestyle factors and survival (Table 2).

In the univariate Cox proportional hazards models, the *ERCC1* rs735482 CC genotype was marginally significantly associated with poor DFS (HR = 1.53, 95% CI = 0.99–2.38; *p* = 0.058). The *XPA* rs10817938 CC genotype was significantly associated with an increased risk of worse OS (HR = 2.97, 95% CI = 1.20–7.35; *p* = 0.019), and DFS (HR = 2.61, 95% CI = 1.06–6.41; *p* = 0.037), respectively (Appendix A).

The results for the multivariate Cox proportional hazards models with covariates adjusted for all selected SPNs are shown in Table 3. Only the *XPC* rs2228000 TT genotype (HR = 1.81, 95% CI = 0.99–3.29, *p* = 0.053) showed an increased risk of poor OS at borderline significance compared with the CC+CT genotypes. The *ERCC5* rs17655 CC (HR = 1.50, 95% CI = 1.01–2.24; *p* = 0.045) and *ERCC1* rs735482 CC (HR = 1.61, 95% CI = 1.04–2.51; *p* = 0.034) genotypes were significantly associated with an increased risk of DFS compared with their counterparts with the GG+GC and AA+AC genotypes, respectively, in the recessive models. The test of LD show that SNPs in *ERCC5* block 1 (rs2094258 and rs1047768; D’ = 0.97, *R*^2^ = 0.19) and block 2 (rs17655 and rs873601; D’ = 0.98, *R*^2^ = 0.89), *ERCC1* block (rs3212986 and rs11615; D’ = 1.00, *R*^2^ = 0.18), *XPC* block (rs2228001 and rs2228000; D’ = 1.00, *R*^2^ = 0.28), and *XPA* block (rs1800975 and rs10817938; D’ = 1.00, *R*^2^ = 0.24) were in LD with each other (Appendix A). Of the haplotype constructed from these blocks, only *XPA* GT haplotype (HR = 0.68, 95% CI = 0.47–0.99; *p* = 0.042) showed statistically significant association with OS (Appendix A).

We further conducted a combination analysis for the *ERCC5* rs17655 and *ERCC1* rs735482 polymorphisms and DSF in patients with OSCC. The Kaplan–Meier curves showed borderline significant differences in DFS among the four genotypes (log-rank test *p* = 0.078) (Figure 1). The multivariate Cox proportional models indicated that patients with the combination of *ERCC5* rs17655 CC and *ERCC1* rs735482 CC genotypes exhibited a higher risk of disease recurrence than those with the combination of *ERCC5* rs17655 GG+GC and *ERCC1* rs735482 AA+AC genotypes (HR = 2.60, 95% CI = 1.11–6.09; *p* = 0.027) (Table 4). However, this gene-gene interaction was not statistically significant.

Table 5 shows the subgroup analysis for the association between significant SNPs and DFS stratified by demographic and clinopathological factors. Results show a significant interaction between *ERCC5* rs17655 polymorphism and perineural invasion on the risk for DFS (interaction *p* = 0.008). The *ERCC5* rs17655 CC genotype individuals with perineural invasion (HR = 2.46, 95% CI = 1.46–4.15; *p* < 0.001) had an increased risk for DFS compared to their counterparts with no perineural invasion. Although a significant interaction between *ERCC1* rs735482 polymorphism and vascular invasion was also observed (interaction *p* < 0.001), the harmful effect of CC genotype on recurrence was not present in any subgroup of vascular invasion.

## 4. Discussion

In this study, we investigated the association between potentially functional SNPs in the NER pathway genes and clinical outcomes in male patients with OSCC treated with CCRT. Our findings suggest that the *XPC* rs2228000 TT genotype was marginally significantly associated with increased risk of death, whereas the *ERCC5* rs17655 CC and *ERCC1* rs735482 CC genotypes were significantly associated with the increased risk of relapse.

The NER pathway plays a major role in DNA repair through the removal of bulky DNA lesions formed by ultraviolet (UV) radiation, environmental mutagens, and other chemotherapeutic agents [23,24]. Studies have revealed that variations in DNA-repair capacity are related to cancer risk and prognosis [25,26]. In addition, SNPs in the NER genes modulate susceptibility to efficacy and survival outcome of the treatment in certain types of cancers [11,12]. Therefore, the same phenomena may be exhibited in patients with OSCC, particularly those undergoing CCRT. Such information may be useful in identifying patients who may benefit from alternative therapies to achieve superior survival and improve quality of life.

The *XPC* gene is a key component of the XPC complex, which plays an important role in the early part of the global genome NER. The corresponding protein plays an important function in damage sensing and DNA binding [27]. SNPs in this gene have been found to affect clinical outcomes in various cancer types [18,28,29]. Li and colleagues observed that the *XPC* rs2228000 TT genotypes were associated with shorter OS than the CC+CT genotype individuals in a study of Japanese gastric cancer patients [30]. Another Chinese study demonstrated that patients with the CC genotype of *XPC* rs2228000 have a borderline significant decreased risk of developing gastric cancer compared with those with the CT+TT genotype [31]. This evidence suggests that the T-allele may have a high susceptibility for poor prognosis. Similarly, in our study, the *XPC* rs2228000 TT genotype shows an increased risk of death compared with the CC+CT genotype. Given the importance of the *XPC* gene in the NER pathway, it is possible that variants of *XPC* alter the DNA repair capacity and thereby affect sensitivity to therapeutic agents. However, the association observed in our study was borderline significant and must be interpreted with caution. Furthermore, large studies may be required to confirm these findings.

We also observed that those with the *ERCC5* rs17655 CC and *ERCC1* rs735482 CC genotypes have an increased risk of relapse compared with individuals with the GG+GC and AA+AC genotypes, respectively. Considered a central component in NER, *ERCC5* encodes a specific DNA endonuclease responsible for excision and repair of UV-induced DNA damage [23]. Evidence has linked *ERCC5* polymorphism to chemotherapeutic response and prognosis of tumors [19,20]. The ERCC5 mRNA expression levels were correlated with cytotoxicity of chemotherapy regiments [32]. Additionally, the rs17655 leads to an amino-acid substitution from histidine to aspartic acid, which may lead to differential interacting abilities, thus, influencing the DNA repair efficacy. Song et al. also observed that *ERCC5* rs17655 polymorphism has a moderately increased risk of recurrence in squamous cell carcinoma of the oropharynx [33]. ERCC1 is also a crucial member of the NER pathway that forms a complex with ERCC4, and together with ERCC5, is responsible for DNA incision [34]. Other studies have reported that *ERCC1* affects the clinical outcome and may serve as a potential biomarker for response to cisplatin-based therapy [35,36]. On the basis of these study results, we speculate that the CC genotypes of *ERCC5* rs17655 and *ERCC1* rs735482 may increase the DNA-repair capacity of cancer cells, leading to increased susceptibility to recurrence. Therefore, if these findings are confirmed by other studies, these SNPs may serve as therapeutic biomarkers for clinical outcome in patients with OSCC who undergo CCRT.

Our study has several limitations. First, the hospital-based nature of the patients may have led to selection bias. Secondly, not all SNPs in the entire NER pathway were used. Some rare functional SNPs may have an influence on survival. Finally, the human papilloma virus (HPV) status and inflammatory cytokines expression of patients was not included in the analysis and may limit the interpretation of our findings; hence, HPV and cytokines may affect survival [37,38]. However, a major strength of our study is that all the patients had a similar tumor stage and received the same treatment. This meant that the effect of different treatments was excluded, which might lead to different levels of DNA damage and repair.

## 5. Conclusions

We investigated the association between key potentially functional SNPs in the NER pathway and susceptibility for death or relapse in male patients with advanced OSCC who were treated with adjuvant CCRT. Our findings showed that the CC genotypes of *ERCC5* rs17655 and *ERCC1* rs735482 were associated with an increased risk of recurrence. CCRT may not be beneficial for these patients; therefore, alternative treatments are required. To our knowledge, this is the largest study to investigate the association between NER polymorphisms and survival in patients with OSCC treated with CCRT in ethnic Chinese. Our findings may require further confirmation in studies with a larger sample size or other ethnic populations.

## Figures and Tables

**Figure 1 jcm-08-00033-f001:**
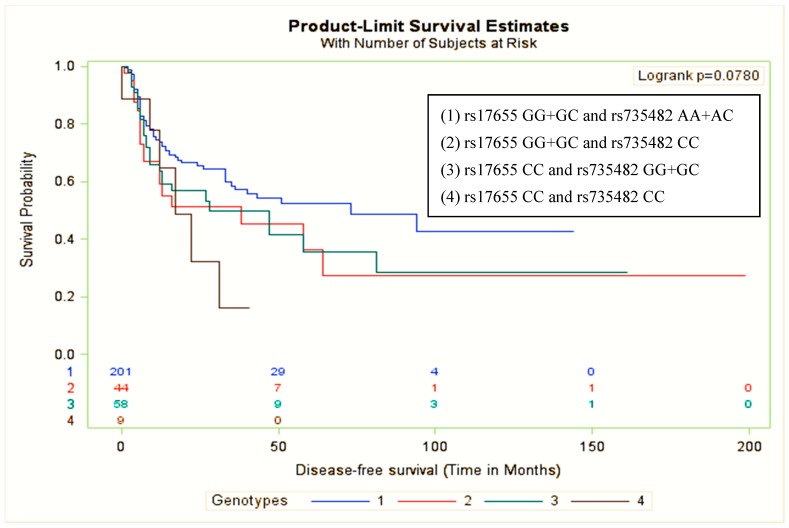
Kaplan–Meier survival curve for the combined *ERCC5* rs17655 and *ERCC1* rs735482 polymorphisms and disease-free survival in patients with oral squamous cell carcinoma treated with concurrent chemoradiotherapy. The figure illustrates a borderline significant difference in recurrence among the four groups (log-rank test *p* = 0.078).

**Table 1 jcm-08-00033-t001:** Demographic and clinical characteristics of patients with oral squamous cell carcinoma (OSCC) treated with concurrent chemoradiotherapy (CCRT).

Variable	*n* (%)
Total	319 (100)
Mean age (SD), years	49.72 (9.8)
Age, years	
<50	164 (51.41)
≥50	155 (48.59)
Ethnicity of father	
Taiwanese	230 (72.10)
Hakka	72 (22.57)
Mainland Chinese	17 (5.33)
Ethnicity of mother	
Taiwanese	239 (74.92)
Hakka	74 (23.20)
Mainland Chinese	6 (1.88)
BMI, kg/m^2^	
<18.5	22 (6.90)
18.5–23.9	157 (49.22)
≥24	140 (43.89)
Cigarette smoking	
Never	47 (14.73)
Ever	272 (85.27)
Alcohol drinking	
Never	98 (30.72)
Ever	221 (69.28)
Betel nut chewing	
Never	44 (13.79)
Ever	275 (86.21)
Tea drinking	
Never	163 (51.10)
Ever	156 (48.90)
Coffee drinking	
Never	243 (76.18)
Ever	76 (23.82)
Tumor differentiation	
Well differentiated	51 (15.99)
Moderate	210 (65.83)
Poor	55 (17.24)
Unclear	3 (0.94)
Primary tumor size	
T1–T2	122 (38.24)
T3–T4	197 (61.76)
Nodal involvement	
N0–N1	102 (31.97)
N2–N3	217 (67.92)
Perineural invasion	
No	142 (44.51)
Yes	177 (55.49)
Vascular invasion	
No	300 (94.04)
Yes	19 (5.96)
Lymphatic invasion	
No	279 (87.46)
Yes	40 (12.54)
Extranodal extension	
No	114 (35.74)
Yes	205 (64.26)
Pathologic TNM stage	
III	42 (13.17)
IV	277 (86.83)

OSCC, oral squamous cell carcinoma; SD, standard deviation; BMI, body mass index.

**Table 2 jcm-08-00033-t002:** Univariate association of demographic and clinical factors with survival in patients with OSCC treated with CCRT.

Variable	Overall Survival		Disease-Free Survival	
	HR (95% CI)	*p* Value	HR (95% CI)	*p* Value
Age				
<50	1.00		1.00	
≥50	0.67 (0.44–1.03)	0.068	0.81 (0.57–1.15)	0.239
Ethnicity of father				
Taiwanese	1.00		1.00	
Hakka	0.88 (0.51–1.50)	0.633	0.66 (0.40–1.08)	0.101
Mainland Chinese	1.27 (0.55–2.93)	0.577	1.72 (0.94–3.14)	0.079
Ethnicity of mother				
Taiwanese	1.00		1.00	
Hakka	0.75 (0.43–1.29)	0.295	0.67 (0.42–1.08)	0.100
Mainland Chinese	1.80 (0.57–5.72)	0.320	1.60 (0.59–4.37)	0.355
BMI, kg/m^2^				
18.5–23.9	1.00		1.00	
<18.5	0.85 (0.36–1.98)	0.705	1.49 (0.79–2.84)	0.223
≥24	0.66 (0.42–1.03)	0.068	0.84 (0.58–1.23)	0.372
Cigarette smoking				
Never	1.00		1.00	
Ever	0.89 (0.51–1.57)	0.675	1.02 (0.62–1.68)	0.938
Alcohol drinking				
Never	1.00		1.00	
Ever	0.99 (0.64–1.54)	0.971	1.10 (0.75–1.61)	0.636
Betel nut chewing				
Never	1.00		1.00	
Ever	1.09 (0.58–2.05)	0.794	1.40 (0.80–2.45)	0.240
Tea drinking				
Never	1.00		1.00	
Ever	0.93 (0.61–1.41)	0.727	1.07 (0.75–1.53)	0.701
Coffee drinking				
Never	1.00		1.00	
Ever	0.65 (0.37–1.13)	0.123	1.27 (0.85–1.89)	0.251
Tumor differentiation				
Well differentiated	1.00		1.00	
Moderate	0.94 (0.55–1.60)	0.819	0.89 (0.56–1.44)	0.645
Poor	0.84 (0.40–1.74)	0.629	1.22 (0.68–2.18)	0.500
Unclear	2.80 (0.37–1.19)	0.319	1.66 (0.22–2.39)	0.621
Primary tumor size				
T1–T2	1.00		1.00	
T3–T4	1.29 (0.83–2.01)	0.258	1.72 (1.17–2.53)	0.006 *
Nodal involvement				
N0–N1	1.00		1.00	
N2–N3	2.41 (1.42–4.10)	0.001 *	1.63 (1.09–2.44)	0.018 *
Perineural invasion				
No	1.00		1.00	
Yes	1.29 (0.85–1.98)	0.238	1.27 (0.89–1.83)	0.189
Vascular invasion				
No	1.00		1.00	
Yes	1.43 (0.66–3.10)	0.366	0.43 (0.43–2.00)	0.856
Lymphatic invasion				
No	1.00		1.00	
Yes	2.18 (1.30–3.67)	0.003 *	1.37 (0.82–2.28)	0.233
Extranodal extension				
No	1.00		1.00	
Yes	3.91 (2.13–7.19)	<0.001 *	1.79 (1.20–2.69)	0.005 *
Pathologic TNM stage				
III	1.00		1.00	
IV	1.64 (0.81–3.29)	0.168	1.66 (0.93–2.96)	0.087

BMI, body mass index; OSCC, oral squamous cell carcinoma; HR, hazard ratio; CI, confidence interval. * *p* < 0.05.

**Table 3 jcm-08-00033-t003:** Multivariate association between nucleotide excision repair (NER) candidate single nucleotide polymorphisms (SNPs) and OSCC survival in patients treated with CCRT.

SNPs	Overall Survival		Disease-Free Survival	
	HR (95% CI) ^a^	*p* Value	HR (95% CI) ^b^	*p* Value
*ERCC5*/*XPG*				
rs2094258				
GG	1.00		1.00	
GA	1.14 (0.72–1.79)	0.574	1.07 (0.73–1.57)	0.726
AA	0.66 (0.30–1.44)	0.294	1.02 (0.57–1.83)	0.946
Additive model	0.91 (0.66–1.25)	0.559	1.03 (0.79–1.34)	0.839
Dominant model	1.03 (0.66–1.59)	0.910	1.06 (0.74–1.52)	0.754
Recessive model	0.61 (0.29–1.28)	0.193	0.99 (0.57–1.70)	0.958
rs1047768				
TT	1.00		1.00	
TC	1.11 (0.72–1.72)	0.635	0.93 (0.64–1.36)	0.715
CC	0.63 (0.25–1.62)	0.339	0.76 (0.39–1.51)	0.438
Additive model	0.94 (0.67–1.31)	0.699	0.90 (0.68–1.19)	0.453
Dominant model	1.03 (0.67–1.58)	0.891	0.90 (0.63–1.29)	0.574
Recessive model	0.60 (0.24–1.49)	0.272	0.79 (0.41–1.53)	0.482
rs17655				
GG	1.00		1.00	
GC	0.95 (0.58–1.54)	0.829	0.86 (0.56–1.31)	0.482
CC	0.93 (0.51–1.69)	0.811	1.38 (0.86–2.19)	0.180
Additive model	0.96 (0.72–1.30)	0.803	1.16 (0.91–1.49)	0.238
Dominant model	0.94 (0.60–1.49)	0.799	1.01 (0.69–1.49)	0.954
Recessive model	0.96 (0.58–1.61)	0.881	1.50 (1.01–2.24)	0.045 *
rs873601				
AA	1.00		1.00	
AG	0.77 (0.47–1.26)	0.298	0.84 (0.55–1.31)	0.448
GG	0.84 (0.47–1.51)	0.556	1.29 (0.80–2.09)	0.301
Additive model	0.91 (0.67–1.23)	0.530	1.14 (0.88–1.48)	0.311
Dominant model	0.79 (0.50–1.26)	0.304	0.97 (0.65–1.46)	0.900
Recessive model	1.00 (0.61–1.64)	0.988	1.44 (0.97–2.14)	0.070
*ERCC2*/*XPD*				
rs13181				
TT	1.00		1.00	
TG	1.03 (0.59–1.81)	0.909	1.00 (0.61–1.64)	0.997
GG	-	0.989	-	0.984
Additive model	1.03 (0.59–1.80)	0.921	1.00 (0.61–1.63)	0.993
Dominant model	1.03 (0.59–1.81)	0.915	1.00 (0.61–1.64)	0.998
Recessive model	-	0.989	-	0.984
rs1799793				
GG	1.00		1.00	
GA	1.10 (0.58–2.07)	0.769	1.01 (0.58–1.73)	0.983
AA	-	0.99	-	-
Additive model	1.10 (0.58–2.06)	0.778	1.01 (0.58–1.73)	0.983
Dominant model	1.10 (0.58–2.07)	0.773	1.01 (0.58–1.73)	0.983
Recessive model	-	0.99	-	-
*ERCC1*				
rs735482				
AA	1.00		1.00	
AC	0.72 (0.46–1.14)	0.163	0.86 (0.57–1.29)	0.466
CC	0.83 (0.44–1.59)	0.580	1.47 (0.89–2.44)	0.134
Additive model	0.86 (0.62–1.19)	0.352	1.16 (0.89–1.52)	0.283
Dominant model	0.75 (0.48–1.15)	0.183	0.98 (0.67–1.44)	0.934
Recessive model	1.01 (0.56–1.83)	0.975	1.61 (1.04–2.51)	0.034 *
rs3212986				
GG	1.00		1.00	
GT	1.20 (0.76–1.89)	0.436	0.95 (0.66–1.39)	0.806
TT	1.04 (0.48–2.28)	0.922	0.96 (0.50–1.85)	0.912
Additive model	1.08 (0.78–1.51)	0.642	0.97 (0.73–1.29)	0.833
Recessive model	1.17 (0.75–1.82)	0.481	0.96 (0.67–1.37)	0.805
Dominant model	0.94 (0.45–1.97)	0.870	0.99 (0.53–1.85)	0.967
rs11615				
CC	1.00		1.00	
CT	1.23 (0.79–1.91)	0.351	0.85 (0.59–1.23)	0.388
TT	0.90 (0.35–2.28)	0.817	0.72 (0.33–1.59)	0.416
Additive model	1.08 (0.77–1.51)	0.661	0.85 (0.63–1.14)	0.282
Dominant model	1.18 (0.77–1.81)	0.438	0.83 (0.58–1.19)	0.316
Recessive model	0.81 (0.33–2.01)	0.652	0.78 (0.36–1.69)	0.525
*XPC*				
rs2228001				
AA	1.00		1.00	
AC	1.07 (0.68–1.69)	0.766	1.21 (0.83–1.78)	0.329
CC	0.72 (0.30–1.73)	0.457	0.75 (0.37–1.54)	0.432
Additive model	0.94 (0.67–1.32)	0.716	0.99 (0.75–1.31)	0.941
Dominant model	1.01 (0.65–1.58)	0.955	1.12 (0.77–1.63)	0.539
Recessive model	0.69 (0.30–1.60)	0.384	0.68 (0.34–1.34)	0.260
rs2228000				
CC	1.00		1.00	
TC	1.06 (0.66–1.68)	0.822	0.81 (0.55–1.18)	0.267
TT	1.86 (0.97–3.56)	0.062	1.11 (0.59–2.08)	0.758
Additive model	1.28 (0.92–1.77)	0.144	0.94 (0.70–1.25)	0.652
Dominant model	1.18 (0.76–1.83)	0.457	0.85 (0.59–1.22)	0.373
Recessive model	1.81 (0.99–3.29)	0.053	1.23 (0.67–2.26)	0.501
*XPA*				
rs1800975				
AA	1.00		1.00	
AG	0.83 (0.50–1.36)	0.461	0.83 (0.51–1.36)	0.462
GG	0.65 (0.35–1.21)	0.175	0.65 (0.35–1.21)	0.174
Additive model	0.81 (0.60–1.10)	0.174	0.90 (0.70–1.16)	0.426
Dominant model	0.77 (0.48–1.24)	0.282	0.77 (0.48–1.24)	0.283
Recessive model	0.74 (0.44–1.25)	0.260	0.73 (0.44–1.25)	0.258
rs10817938				
TT	1.00		1.00	
TC	1.08 (0.68–1.72)	0.731	1.28 (0.87–1.86)	0.209
CC	1.68 (0.65–4.33)	0.286	1.88 (0.75–4.74)	0.180
Additive model	1.18 (0.82–1.69)	0.386	1.31 (0.95–1.80)	0.095
Dominant model	1.15 (0.74–1.77)	0.543	1.32 (0.91–1.90)	0.139
Recessive model	1.64 (0.64–4.17)	0.304	1.73 (0.69–4.32)	0.239

OSCC, oral squamous cell carcinoma; SNPs, single nucleotide polymorphisms; HR, hazard ratio; CI, confidence interval. ^a^ Adjusted for age, BMI, N stage, lymphatic invasion, and extranodal extension. ^b^ Adjusted for age, T stage, N stage, and extranodal extension. * *p* <0.05.

**Table 4 jcm-08-00033-t004:** Interaction between the *ERCC5* rs17655 and *ERCC1* rs735482 polymorphisms on the disease-free survival of patients with OSCC treated with CCRT.

*ERCC5* rs17655	*ERCC1* rs735482	No.	Event	HR (95% CI) ^a^	*p* Value
GG+GC	AA+AC	206	71	1.00	
GG+GC	CC	45	19	1.63 (0.98–2.72)	0.060
CC	GG+GC	59	29	1.52 (0.98–2.37)	0.062
CC	CC	9	6	2.60 (1.11–6.09)	0.027 *
*p* for interaction				0.929	

OSCC, oral squamous cell carcinoma; HR, hazard ratio; CI, confidence interval. ^a^ Adjusted for age, T stage, N stage, and extranodal extension. * *p* < 0.05.

**Table 5 jcm-08-00033-t005:** Association between the *ERCC5* rs17655 and *ERCC1* rs735482 polymorphisms and the disease-free survival in OSCC patients treated with CCRT stratified by demographic and clinopathological factors.

	*ERCC5* rs17655			*ERCC1* rs735482		
Variable	HR (95% CI) ^a^	*p* Value	*p* _Interaction_	HR (95% CI) ^a^	*p* Value	*p* _Interaction_
Age			0.486			0.078
<50	1.74 (1.00–3.03)	0.051		1.04 (0.53–2.06)	0.911	
≥50	1.24 (0.69–2.23)	0.473		2.76 (1.47–5.19)	0.002 *	
BMI, kg/m^2^			0.160			0.597
18.5–23.9	1.64 (0.95–2.82)	0.075		1.89 (1.04–3.44)	0.037 *	
<18.5	3.06 (0.65–14.49)	0.159		1.30 (0.05–31.65)	0.871	
≥24	0.81 (0.37–1.77)	0.601		1.30 (0.62–2.72)	0.495	
Cigarette smoking			0.664			0.975
Never	1.03 (0.35–2.98)	0.962		1.99 (0.52–7.58)	0.316	
Ever	1.54 (0.99–2.41)	0.056		1.61 (1.00–2.59)	0.051	
Alcohol drinking			0.993			0.676
Never	1.59 (0.74–3.42)	0.240		1.43 (0.64–3.22)	0.387	
Ever	1.52 (0.94–2.46)	0.091		1.72 (1.00–2.96)	0.050	
Betel nut chewing			0.445			0.968
Never	2.06 (0.53–8.11)	0.299		1.64 (0.34–7.86)	0.536	
Ever	1.41 (0.92–2.14)	0.112		1.63 (1.02–2.62)	0.043 *	
Tea drinking			0.331			0.986
Never	1.88 (1.05–3.35)	0.033 *		1.60 (0.87–2.94)	0.134	
Ever	1.27 (0.73–2.24)	0.400		1.57 (0.81–3.04)	0.183	
Coffee drinking			0.073			0.077
Never	1.30 (0.82–2.08)	0.066		2.09 (1.28–3.41)	0.003 *	
Ever	2.83 (1.27–6.33)	0.011 *		0.72 (0.22–2.42)	0.601	
Tumor differentiation			0.492			0.949
Well differentiated	1.08 (0.43–2.68)	0.875		3.74 (1.26–11.09)	0.018 *	
Moderate	1.43 (0.84–2.42)	0.184		1.06 (0.57–1.98)	0.858	
Poor	2.15 (0.88–5.22)	0.093		3.48 (1.23–9.88)	0.019 *	
Primary tumor size			0.146			0.394
T1–T2	0.96 (0.45–2.03)	0.905		1.12 (0.49–2.56)	0.788	
T3–T4	1.83 (1.14–2.93)	0.013 *		1.76 (1.04–3.00)	0.036 *	
Nodal involvement			0.328			0.125
N0–N1	2.25 (1.06–4.81)	0.036 *		3.16 (1.47–6.80)	0.003 *	
N2–N3	1.32 (0.82–2.13)	0.252		1.23 (0.69–2.19)	0.479	
Perineural invasion			0.008 *			0.416
No	0.77 (0.40–1.49)	0.429		2.07 (1.06–4.03)	0.032 *	
Yes	2.46 (1.46–4.15)	<0.001 *		1.32 (0.72–2.44)	0.370	
Vascular invasion			0.410			<0.001 *
No	1.45 (0.96–2.19)	0.078		1.51 (0.96–2.37)	0.076	
Yes	1.27 (0.16–10.34)	0.826		-		
Lymphatic invasion			0.553			0.449
No	1.43 (0.92–2.23)	0.111		1.71 (1.07–2.73)	0.025 *	
Yes	1.84 (0.64–5.31)	0.260		0.90 (0.20–4.16)	0.892	
Extranodal extension			0.253			0.720
No	2.35 (1.12–4.94)	0.024 *		2.17 (0.87–5.40)	0.097	
Yes	1.31 (0.81–2.13)	0.277		1.52 (0.90–2.57)	0.118	
Pathologic TNM stage			0.850			0.585
III	1.70 (0.48–6.09)	0.414		4.82 (0.98–23.58)	0.053	
IV	1.51 (0.99–2.31)	0.055		1.54 (0.96–2.47)	0.075	

OSCC, oral squamous cell carcinoma; HR, hazard ratio; CI, confidence interval; Int, interaction. ^a^ Adjusted for age, T stage, N stage, and extranodal extension. * *p* < 0.05.

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
