# Peer review of "Polymorphisms in ERCC5 rs17655 and ERCC1 rs735482 Genes Associated with the Survival of Male Patients with Postoperative Oral Squamous Cell Carcinoma Treated with Adjuvant Concurrent Chemoradiotherapy"

_jcm, 2019, doi:10.3390/jcm8010033_

Reviewer 1 Report

The manuscript by Senghore et al identify the single nucleotide polymorphisms in nucleotide excision repair (NER) pathway genes   and clinical outcomes of patients with oral squamous cell carcinoma treated with concurrent chemoradiotherapy. The manuscript is well written and conclusive of findings. However, the authors should address the following comments to strengthen the manuscript.    

1. Clarify the study subjects in the methods section ie male participants, both maternal and paternal subjected employed in the study. Accordingly, they should clarify the title should specify the study subjects used.  

2. They should address diagnosis criteria used may not be homogeneous among studies included.

3. It is unclear if the polymorphisms appears in the recurrent state vs primary tumors, location, tumor size, stage and invasion potential.

4. The authors should include clinical outcome on the course of radiation therapy (RT) aside from the median dose ie., +/-surgery,  +/- concurrent chemo with RT etc.

5. The authors should provide the proteins encoding and potential function of NER pathway related genes ERCC5, ERCC1, ERCC2, XPC, and XPA selected for the study.

6. They have noted the potential role of papillomavirus in sursvival as limitation.  However, they should also discuss other survival/growth factors such as cytokines related to OSCC.

Author Response

Responses to Reviewers’ Comments

Paper No: jcm-405120

Titled: Polymorphisms in NER genes were associated with the survival of patients with postoperative oral squamous cell carcinoma treated with adjuvant concurrent chemoradiotherapy

New title: Polymorphisms in ERCC5 rs17655 and ERCC1 rs735482 genes were associated with the survival of male patients with postoperative oral squamous cell carcinoma treated with adjuvant concurrent chemoradiotherapy

Response to Reviewer 1

Comments and Suggestions for Authors

The manuscript by Senghore et al identify the single nucleotide polymorphisms in nucleotide excision repair (NER) pathway genes  and clinical outcomes of patients with oral squamous cell carcinoma treated with concurrent chemoradiotherapy. The manuscript is well written and conclusive of findings.   However, the authors should address the following comments to strengthen the manuscript.   

Reply:

Thank you for your time and constructive criticisms.

1. Clarify the study subjects in the methods section ie male participants, both maternal and paternal subjected employed in the study. Accordingly, they should clarify the title should specify the study subjects used. 

Reply:

In the methods section, we had described the characteristics of subjects used in the study. Accordingly, the title has been modified to specify the subjects used. In addition, as suggested by reviewer 2, we have specified in the title the polymorphisms that were found to associate with survival.

2. They should address diagnosis criteria used may not be homogeneous among studies included.

Reply:

All the patients in our study were diagnosed using the seventh edition of the American Joint Committee on Cancer – TNM staging system, thus showing homogeneity of diagnosis. A detail description of this can be found in the methodology section 2.2. In addition, we have also specified this in section 2.1, Study population on page 5, line 86.

3. It is unclear if the polymorphisms appears in the recurrent state vs primary tumors, location, tumor size, stage and invasion potential.

Reply:

The polymorphisms in our study were identified from germline DNA extracted from venous blood samples. The information has now been specified in the methodology, page 6, section 2.2, lines 115 and 116. In addition, we have conducted genotype frequency distribution analysis to show the distribution of genotypes in terms of clinical factors. This information is shown in Table S1 and results section page 10, lines 183 – 188.

4. The authors should include clinical outcome on the course of radiation therapy (RT) aside from the median dose ie., +/-surgery,  +/- concurrent chemo with RT etc.

Reply:

All the patients in our study first received radical surgery followed by concurrent chemoradiation therapy.  Accordingly, we did not categorize the study group into RT +/- surgery , +/- CCRT as per reviewer's suggestion. In addition, the endpoint of our evaluation of clinical outcome was recurrence of disease and death. We analyzed and presented them as overall and disease-free survival. Some investigators evaluated the treatment related complications such as leukopenia or mucositis as the clinical outcome. However, these complications were not recorded in our database and further analysis is not possible.

5. The authors should provide the proteins encoding and potential function of NER pathway related genes ERCC5, ERCC1, ERCC2, XPC, and XPA selected for the study.

Reply:

In line with the above comment, we have added a paragraph in the introduction that describes the function of the proteins for the selected genes. See the introduction section page 3, lines 67 – 75.

6. They have noted the potential role of papillomavirus in survival as limitation.  However, they should also discuss other survival/growth factors such as cytokines related to OSCC.

Reply:

We are aware of the potential effect that cytokines may have on the progression of tumors cells. However, this information was not evaluated in our study and is included as a limitation. See the discussion section, page 23, lines 321 – 324.

Reviewer 2 Report

In this manuscript, the Authors investigated a set of 13 SNPs in 5 genes involved in nucleotide excision repair (NER), in particular in ERCC5, ERCC1, ERCC2, XPC and XPA, in a cohort made of 319 oral squamous cell carcinomas. The manuscript is well written however it is not really innovative as recently previous reports evaluated similar markers in advanced OSCC: see for instance

-       Loper-Aguiar L et al Oncotarget. 2017 Mar 7;8(10):16190-16201

-       Nanda SS et al . Int J Radiat Oncol Biol Phys. 2018 Jul 1;101(3):593-601

-       Gao C PLoS One. 2016 Sep 13;11(9):e0160801

Additionally, the title indicates a close association among some NER alleles and the survival of OSCC patients treated with adjuvant chemoradiotherapy, but data shown here pointed out a weak association i.e: in univariate analysis only rs10817938 showed p: 0.019 for risk of worse OS with only 5 events. In multivariate analysis only rs17655 and rs735482 were associated with an increased risk of DFS (P: 0.045 and P: 0.034 respectively). Probably the title should be changed highlighting the informative alleles above mentioned.

This manuscript suffers from the following major points: 

-      Section Results, lane 171: the median follow-up is very short (15 months), it should be advisable to have a longer median follow up of at least 2 years or more in order to have consistent results

-      Since fresh-frozen material from tumor and normal adjacent non-tumor sample of these 319 cases was available, The authors could evaluate the same set of SNPs in tumor and matched normal samples to identify loss of heterozygosity/allelic imbalance, to give novelty to this paper and to verify if NER genes will reveal loss of function in OSCC?

-      Linkage Disequilibrium (LD) analysis was not done. It’s important to do it considering all of 13 SNPs evaluated.

-      The survival analysis could be improved: subgroups should be created and analyzed separately for SNPs stratification with the same TNM and as follows: smokers; chewed betel nut; drank alcohol; perineural invasion; lymphatic invasion. This may render homogeneous groups of patients with similar clinical features and may outlined the power of alleles limiting the effect of staging

-       HPV DNA detection was not evaluated. It affects the prognosis so it should also be included

Author Response

Response to Reviewer 2

Reviewer's report:

In this manuscript, the Authors investigated a set of 13 SNPs in 5 genes involved in nucleotide excision repair (NER), in particular in ERCC5, ERCC1, ERCC2, XPC and XPA, in a cohort made of 319 oral squamous cell carcinomas. The manuscript is well written however it is not really innovative as recently previous reports evaluated similar markers in advanced OSCC: see for instance

-       Loper-Aguiar L et al Oncotarget. 2017 Mar 7;8(10):16190-16201

-       Nanda SS et al . Int J Radiat Oncol Biol Phys. 2018 Jul 1;101(3):593-601

-       Gao C PLoS One. 2016 Sep 13;11(9):e0160801

Additionally, the title indicates a close association among some NER alleles and the survival of OSCC patients treated with adjuvant chemoradiotherapy, but data shown here pointed out a weak association i.e: in univariate analysis only rs10817938 showed p: 0.019 for risk of worse OS with only 5 events. In multivariate analysis only rs17655 and rs735482 were associated with an increased risk of DFS (P: 0.045 and P: 0.034 respectively). Probably the title should be changed highlighting the informative alleles above mentioned.

Reply:

Thank you for your time and constructive criticisms. The title is now modified to specify the SNPs that were found to associate with survival. In addition, we have cited the above-mentioned references and mention in the discussion the weak association as a limitation to our result. See page 22 lines 297 – 299.

This manuscript suffers from the following major points: 

-      Section Results, lane 171: the median follow-up is very short (15 months), it should be advisable to have a longer median follow up of at least 2 years or more in order to have consistent results

Reply:

We are aware that having a longer median follow-up time of more than 24 months is more desirable to show consistent results. However, give the advance stage (III and IV) and poor clinical characteristic of our sample it would be difficult to achieve such a median follow-up time. As shown, despite the long follow-up time of 199 months, the median follow-up was 15 and 12 months for overall and disease-free survival, respectively. From our literature search, our study is one of the longest follow-ups of OSCC patients study.  Similar studies on oral cancer patients show shorter follow-up times (about 24 to 36 months) and median follow-up times of less than 24 months [1-3].

References

1.          Lopes-Aguiar, L.; Costa, E.F.D.; Nogueira, G.A.S., et al. XPD c.934G>A polymorphism of nucleotide excision repair pathway in outcome of head and neck squamous cell carcinoma patients treated with cisplatin chemoradiation. Oncotarget 2016, 8, 16190-16201.

2.          Nanda, S.S.; Gandhi, A.K.; Rastogi, M., et al. Evaluation of XRCC1 Gene Polymorphism as a Biomarker in Head and Neck Cancer Patients Undergoing Chemoradiation Therapy. Int. J.Radiat. Oncol. Biol. Phys. 2018, 101, 593-601.

3.          Gao, C.; Wang, J.; Li, C., et al. A Functional Polymorphism (rs10817938) in the XPA Promoter Region Is Associated with Poor Prognosis of Oral Squamous Cell Carcinoma in a Chinese Han Population. PLoS One 2016, 11, e0160801.

-      Since fresh-frozen material from tumor and normal adjacent non-tumor sample of these 319 cases was available, The authors could evaluate the same set of SNPs in tumor and matched normal samples to identify loss of heterozygosity/allelic imbalance, to give novelty to this paper and to verify if NER genes will reveal loss of function in OSCC?

Reply:

Thank you for this comment that could add novelty to our study, however adding this aspect to our study will require ethical approval as the current ethical approval does not include this aspect of the study. We have taken note of it and will consider this in our future studies.

-      Linkage Disequilibrium (LD) analysis was not done. It’s important to do it considering all of 13 SNPs evaluated.

Reply:

We have now conducted linkage disequilibrium and haplotype analysis including assessing the effects of haplotypes on survival. This information is incorporated in the methodology section page 8, lines 167 – 170, and results section page 14 - 15 lines 217 – 223. The results are also submitted as Supplementary material, Figure S1 and Table S3.

-      The survival analysis could be improved: subgroups should be created and analyzed separately for SNPs stratification with the same TNM and as follows: smokers; chewed betel nut; drank alcohol; perineural invasion; lymphatic invasion. This may render homogeneous groups of patients with similar clinical features and may outlined the power of alleles limiting the effect of staging

Reply:

We have done subgroup analysis and the results have been incorporated in the manuscript. See table 5 of the results and page 19, lines 250 – 258. In addition, we have conducted genotype distribution analysis for the clinical factors. See results section page 10 lines 182 – 188 and supplementary material, Table S1. 

-       HPV DNA detection was not evaluated. It affects the prognosis so it should also be included

Reply:

We are aware of the possible effect that HPV infection could have on OSCC risk and clinical outcome. This information was not available in our database. However, we are of the view that inclusion of this information will not significantly alter our results given that HPV status in oral tumors does not significantly impact treatment response and survival. Somatic mutations are rare in HPV positive tumors, especially type 16. In addition, our entire patient received CCRT, which can lead to DNA damage that could be repaired by several pathways including the p53 pathway. Evidence has shown that Head and Neck cancer patient who received chemo and radiotherapy had improved clinical outcome regardless of HPV status [1-3].  Therefore, HPV status may have a limited effect on our patients’ survival. Despite this, we have included it as a limitation. See page 23, lines 321 – 324.

References

1.    Rosenthal, D.I.; Harari, P.M.; Giralt, J., et al. Association of Human Papillomavirus and p16 Status With Outcomes in the IMCL-9815 Phase III Registration Trial for Patients With Locoregionally Advanced Oropharyngeal Squamous Cell Carcinoma of the Head and Neck Treated With Radiotherapy With or Without Cetuximab. J Clin Oncol 2016, 34, 1300-1308.

2.    Vermorken, J.B.; Psyrri, A.; Mesia, R.; et al. Impact of tumor HPV status on outcome in patients with recurrent and/or metastatic squamous cell carcinoma of the head and neck receiving chemotherapy with or without cetuximab: retrospective analysis of the phase III EXTREME trial. Ann. Oncol. 2014, 25, 801-807.

3.    Bonner, J.A.; Mesia, R.; Giralt, J.; et al. p16, HPV, and Cetuximab: What Is the Evidence? Oncologist 2017, 22, 811-822.

Round  2

Reviewer 1 Report

None

Reviewer 2 Report

Minor revisions:

Section 2.2: germline DNA can be obtained only from spermatocytes and oocytes, and buffer-coat does not exist so please modify the sentence: 

"Germline DNA was extracted from buffer-coated cells using the standard phenol–chloroform method and prepared for genotyping."

with "DNA was extracted from buffy coat using the standard phenol–chloroform method and prepared for genotyping".

Author Response

The sentence has been modified in the manuscript.